# Experimental Investigations of Forward and Reverse Combustion for Increasing Oil Recovery of a Real Oil Field

**Aysylu Askarova [1],\*, Evgeny Popov [1], Matthew Ursenbach [2], Gordon Moore [2], Sudarshan Mehta [2] and Alexey Cheremisin [1]**

[1] Skolkovo Institute of Science and Technology, Center for Hydrocarbon Recovery, Sikorsky Street 11, 121205 Moscow, Russia; E.Popov@skoltech.ru (E.P.); A.Cheremisin@skoltech.ru (A.C.)
[2] Department of Chemical and Petroleum Engineering, University of Calgary, Calgary, AB T2N 1N4, Canada; ursenbac@ucalgary.ca (M.U.); moore@ucalgary.ca (G.M.); mehta@ucalgary.ca (S.M.)
**\*** Correspondence: aysylu.askarova@skolkovotech.ru

**Abstract:** The work presented herein is devoted to a unique set of forward and reverse combustion tube (CT) experiments to access the suitability and potential of the in situ combustion (ISC) method for the light oil carbonate reservoir. One forward and one reverse combustion tube tests were carried out using the high-pressure combustion tube (HPCT) experimental setup. However, during reverse combustion, the front moved in the opposite direction to the airflow. The results obtained from experiments such as fuel/air requirements, H/C ratio, and recovery efficiency are crucial for further validation of the numerical model. A quantitative assessment of the potential for the combustion was carried out. The oil recovery of forward combustion was as high as 91.4% of the initial oil in place, while that for the reverse combustion test demonstrated a 43% recovery. In the given conditions, forward combustion demonstrated significantly higher efficiency. However, the stabilized combustion front propagation and produced gases of reverse combustion prove its possible applicability. Currently, there is a limited amount of available studies on reverse combustion and a lack of publications within the last decades despite advances in technologies. However, reverse combustion might have advantages over forward combustion for heavy oil reservoirs with lower permeability or might serve as a reservoir preheating technique. These experiments give the opportunity to build and validate the numerical models of forward and reverse combustion conducted at reservoir conditions and test their field application using different scenarios.

**Keywords:** enhanced oil recovery; forward combustion; reverse combustion; high-pressure combustion tube; carbonate reservoir

## 1. Introduction

Potentially, in situ combustion (ISC) can be the most effective enhanced oil recovery (EOR) method for enhancing recoveries from oil fields [1–3]. ISC, in comparison with other EOR processes, has been tested on a very broad range of conditions [4–6]. During this method, energy and material transport is provided through porous media. The heat and combustion gases reduce the viscosity of the driven oil, which subsequently increases its mobility. Two types of combustion have been distinguished, depending on the movement of the hot front: forward and reverse combustion [4,7].

During forward combustion, the combustion zone moves towards the production well in the same direction as airflow [8]. In the first step, the crude oil near the wellbore needs to be ignited using gas burners, electrical heaters, using chemical agents or steam injections. After that, continuous air injection leads to the movement of the combustion front towards the production wells. Ideally,

it results in the almost complete removal of all reservoir liquids from the swept zone (leaving behind a hot, clean rock) [9]. Sometimes, ignition can happen spontaneously if the temperature in the reservoir is high enough and the oil is reasonably reactive [10,11].

Reverse combustion is a method that also can be used for the production of oil that has high viscosity. Air is introduced into the underground formation via an injection well. In one or more adjacent production wells, the mixture of air and hydrocarbons is ignited. The created combustion zone moves away toward the injection well, with the opposite direction to the airflow [4,7,12–14]. Two dependent parameters that are important in terms of defining the progress of the combustion process are the maximum temperature achieved within the combustion zone and the front velocity [15,16]. They can provide insights about recovery efficiency, oil, and gas production rates, and average air requirements.

A one-dimensional flow in a homogenous system is assumed to visualize the reverse combustion [9]. When the steady-state conditions are achieved, the combustion zone recedes at a constant velocity towards the air supply point. The zone moves due to heat conducted through the rock towards the incoming air. The theory behind the reverse combustion is presented in a paper [16] which provides heat, oxygen equations, and a steady-state approximation that are more specific than the flame theory.

Unfortunately, this method is commonly hard to apply and economically unattractive. Firstly, this is because the unreacted oxygen, which is contained in hot produced fluids, requires protection against high temperatures and corrosion (high cost). At the same time, it requires more oxygen, thus increasing the cost. Secondly, it is very hard to achieve significant oil production even in carefully controlled laboratory experiments as, at some point, the reverse combustion would revert to forward [12]. However, the successful application of reverse combustion is possible with desired air permeability, oil saturation, and a sufficient rate of the reaction [7,9]. It might be applicable for very heavy oils American Petroleum Institute (API) gravity 5–20 °API [17] at very low reservoir temperatures due to slow spontaneous ignition up to several years [12]. Another favorable condition for reverse combustion can be low effective permeability, which helps to minimize the reservoir plugging by the mobilized fluids [4]. Reverse combustion plays an important role in coal and tar sands since it can develop highly permeable paths between production and injection wells where, at the second stage forward, combustion can be used [18].

The complex process with multiple oxidation reactions with irregular transitions, occurring during the in situ combustion process is not yet well understood. Meanwhile, a very limited amount of open-published literature is available for reverse combustion with limited insights on the process. The paper by Reed and Tracht [9] discusses uncertainties related to the temperature redistribution near the wall due to the short length of their reactor in late 1960. The deficiency in experimental data is another issue that authors have faced. Different conditions under which reverse combustion becomes attractive must be studied as well as its application to the conventional as well as heavy oil fields [9]. Laboratory experiments on reverse combustion also been conducted in 1960, demonstrating a much lower oil recovery value (50% of OOIP) compared to that for forward combustion (85–95%). The higher amount of oil consumed as fuel during reverse combustion and the front velocity was relatively slower [7].

Another one-dimensional laboratory study was described in the paper [19] in 1985 that was conducted to evaluate the reverse combustion applications. Reverse combustion served as an effective preheating method for the tar sand and the development of plugging was avoided. It was possible to increase the combustion temperatures with air flux alterations. Due to the coking of the remaining bitumen, a very small amount of oil was available for steam-flooding recovery. Low-temperature oxidation (LTO) reactions were dominant which resulted in a low level of carbon oxides, increased H/C ratio, increased water production, and increased oxygen in the product. The produced oil had increased API gravity values and significantly decreased viscosity [19]. Preheating treatment can be beneficial in the cases when the oil saturation is sufficiently high and the effective permeability is low to avoid the reservoir plugging. For example, reverse combustion was applied to the tar sands of the Orinoco deposit and the Athabasca [14].

The paper by Lasaki [20] performs a field case numerical study of in situ reverse combustion and steam flooding. Experiments were also conducted on oil sands. In this case, reverse combustion shifted to forward mode and served as a preheating procedure before steam injection. The reported oil recovery by reverse combustion was in the range of 2–5% original oil-in-place (OOIP), but recovery was accelerated further. Stable reverse combustion can be achieved by a high-communication path or, for example, in fractures [20].

There are few drawbacks of reverse combustion that limit its application. The first is the spontaneous ignition probability [21]. In case of spontaneous ignition near the injector, the oxygen would be consumed and the process would revert to forward combustion. To avoid the spontaneous ignition near the injector, the reservoir should be preheated before air injection [22–24]. Secondly, reverse combustion can be often unstable with narrow combustion channels and, as a result, burns poorly [18,25,26]. However, the main issue is the lack of new experimental studies on reverse combustion within the last few decades, despite the improvements in the technologies.

The main problem with reverse combustion is the limited available experimental data and these experiments were not performed at high pressures and reservoir conditions. The priorities of the combustion tube experiment were to preserve the pressure, temperature, and chemistry (i.e., mineralogy) of the reservoir system being tested. The purpose of this research is to assess the suitability and potential of oil from the subject reservoir for the implementation of an air injection-based enhanced oil recovery (EOR) process in the conditions recreating the real field.

The paper introduces the unique experimental tests of forward and reverse in situ combustion to estimate the overall burning characteristics of the target reservoir restored state core at reservoir pressure 27.2 MPa and reservoir temperature 100 °C, to mimic the conditions that would be encountered in the field. The parameters, such as incremental oil production, air, and fuel requirements, should be measured for a preliminary economic assessment of a field project in the forward and reverse combustion modes. The measure of produced gas compositions, produced oil and water properties gives the benchmark to monitor future field operations.

## 2. Experimental Section

### 2.1. Materials and Methods

Experimental forward and reverse combustion tube tests were conducted using restored state core samples from the actual light carbonate oil field. The core materials were prepared by cleaning the core in the modified Soxhlet-type extractor, dried, and fired overnight in an oven at 350 °C to remove the residual hydrocarbons. The clean cores were crushed to sand-like particle size and sieved to remove fine material. The oil sample with 30 °C API for the test was also selected from the reservoir and centrifuged to remove water.

Figure 1 below demonstrates a schematic diagram of a high-pressure combustion tube (HPCT) experimental setup with its associated equipment where forward and reverse combustion tests were conducted. A detailed representation of the similar combustion tube system as its basic components are provided in the literature [27–29]. The tube was oriented vertically inside the vessel to minimize the effect of gravity. The view of the combustion tube can be found in [27], which includes the sand packed in a thin-walled tube with a 100-mm diameter, thermal insulation to provide uniform heating, heater support column, and electrical heaters. Additionally, the combustion tube was equipped with thermocouples in the center of the sand pack, as well as wall thermocouples. The system of heaters and thermocouples are designed to avoid the radial heat loss and to ensure that the process is not driven by heater regimes. On completion of the packing operation, the tube was sealed, insulated, and inserted into the high-pressure jacket. The tests were carried out on the same experimental setup and under the same conditions, except for the fact that the combustion zone moved in the opposite direction to the airflow during the reverse combustion test. The thermal effect is believed to be the dominant

mechanism during air-injection-based methods instead of the flue gas flooding or gravity segregation during the steam flooding [20,30] The input parameters of both experiments are presented in Table 1.

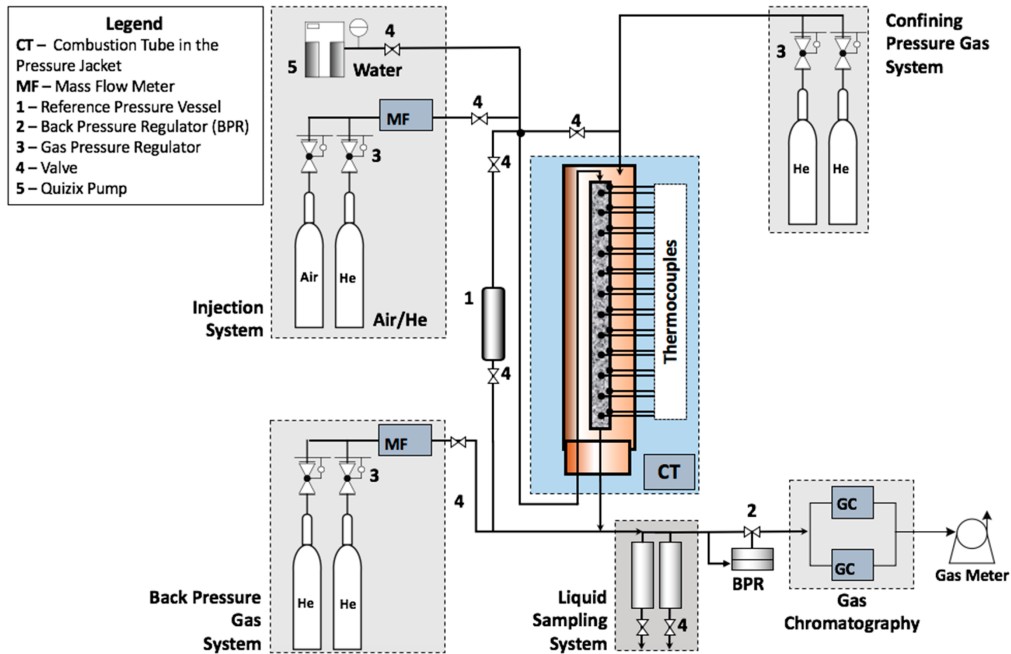

**Figure 1.** High-Pressure Combustion Tube experimental setup.

**Table 1.** HPCT tests input parameters.

|  | **Forward Combustion** |  | **Reverse Combustion** |  |
| --- | --- | --- | --- | --- |
| Number of zones | 33 |  |  |  |
| Tube diameter | 100 mm |  |  |  |
| Pressure | 27.2 MPa |  |  |  |
| Air Injection Flux | 40.4 $m^3$(ST)/$m^2$ h |  |  |  |
| Ignition Temperature | 175 °C |  |  |  |
| API | 30 |  |  |  |
| Porosity | 45.4% |  | 43.2% |  |
| Permeability | 33.6 Darcy |  | 19.5 Darcy |  |
| Reservoir Temperature | 100 °C |  |  |  |
| The average molecular weight of original oil | 217 g/mol |  |  |  |
|  | Before Pressure Up | At the start of Air Injection | Before Pressure Up | At the start of Air Injection |
| So | 70.3 | 66.5 | 71.3 | 69.0 |
| Sw | 29.7 | 33.5 | 28.7 | 31.0 |
| Asphaltenes mass frac | 11.93% |  |  |  |
| Sulfur content | 0.47 |  |  |  |
| H/C ratio | 1.81 |  |  |  |
| Air injection | Top-down |  | Bottom-up |  |
| Oil viscosity | 89/55/27 mPa·s at 15 °C/25 °C/40 °C |  |  |  |
| Original oil density | 0.8795/0.8725/0.8615 g/$cm^3$ at 15 °C/25 °C/40 °C |  |  |  |
| Time of helium purge: hours after the start of air injection | 8.8 |  | 5.5 |  |

The system was maintained in near-adiabatic mode by means of heat supplied externally in such a way that the radial temperature gradient in any plane normal to the axis of the tube approaches zero. Initially, the tube was at reservoir pressure and temperature, except at one end where it was heated to a predetermined "ignition" temperature. When the prescribed ignition temperature was achieved, the air was injected from the "cold" end of the tube (in the case of the reverse combustion test). As the oxygen in the air stream contacted the hot oil, a localized exothermic reaction occurred. The generated heat was conducted and convected away from the reaction zone so that definite temperature and concentration profiles were rapidly established and moved uniformly in the direction opposite of that of airflow.

The objective of the dry forward and reverse combustion tube experiments using a 100-mm HPCT system was to investigate and compare the in situ combustion behavior in the two different process configurations. The tests were performed at a pressure of 27.2 MPa (4000 psia) using synthetic air (21.28-mole% oxygen, the balance being nitrogen) at an air injection flux of 40.4 m$^3$(ST)/m$^2$ h at an ignition temperature 175 °C.

This regime allowed the collection of the requisite amount of produced gas composition and the following detailed analysis. Combustion tube tests give such parameters as an equivalent hydrogen–carbon ratio (H/C) and an idea of the stoichiometry for the high-temperature process.

These experiments were conducted to obtain the information regarding the stoichiometry and implement field design parameters, analyze the combustion front, product gas composition, and temperature profiles.

### 2.2. Forward Combustion

HPCT equipment is presented in Figure 2. The core holder was oriented vertically, the air was injected top-down, and fluids produced during the experiment were collected at the bottom of the tube.

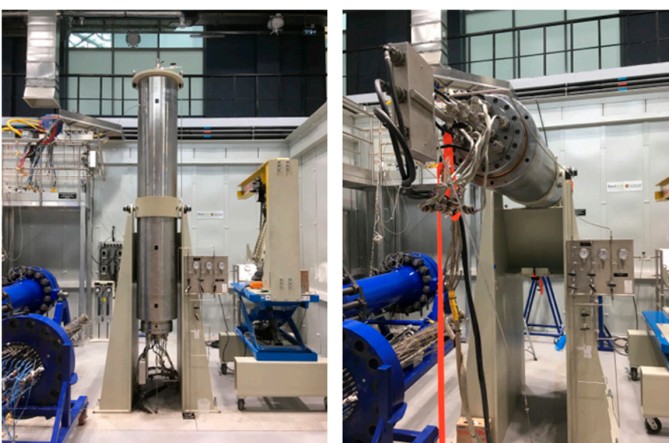

**Figure 2.** High-pressure combustion tube (HPCT) Equipment Images.

Water was initially injected at 1.0 mL/min to pressurize the system and was terminated shortly after the start of air injection. The core was preheated to the reservoir temperature of 100 °C. One hour before the start of air injection first three Zones (first 15 cm) of the combustion tube were commenced to heat up to the ignition temperature of 175 °C, and when this ignition temperature was reached, synthetic air was injected into the top inlet of the core. The first sign of ignition was observed 40 min after the start of air injection.

Temperatures in Zones 2 and 3 started to increase, while the combustion tube was on adiabatic control where wall temperatures were set to track the core temperatures within 5 °C to minimize heat losses. It should be mentioned that no helium was injected through the core before the injection of air.

### 2.3. Reverse Combustion

In the case of reverse combustion, the core holder was also oriented vertically, but in contrast to forward combustion in the given case, the air was injected from the bottom up. Ignition and production were carried out from the top of the core pack since the test is operated in the reverse combustion mode.

Water was also injected with the same flux and the core is gradually preheated to reservoir temperature, then the first three zones were set to the ignition temperature similar to Section 2.1. Due to pressure fluctuations, the air injection rate is increased and decreased several times during the first hour and stabilized at the above-noted designed injection rate. During this unstable period, air reached the top end of the core where ignition zones were located. Wall temperatures were set to "near" adiabatic

control to track the core temperatures within 5 °C as in forward combustion experiment. There was no He injection before air injection.

The reverse combustion front advanced downward through the core opposite to the direction of the injected airflow. At 5.6 h after the start of air injection, the leading edge of the high-temperature front reached Zone 30, air injection was terminated and He was injected at the same rate as the air. Wall heaters were not turned off when the helium purge was initiated, enabling the continuation of the burning process by consuming part of the air that was stored ahead of the combustion front. The helium injection continued for 7.38 h and then the system was bled down. Liquid production was intermittently collected for later analyses; additionally, online gas composition analysis was carried out.

## 3. Results

### 3.1. Forward Combustion

The centerline temperature profiles for each zone of the combustion tube are presented in Figure 3. Heating of the three ignition zones starting at −0.9 h can be observed.

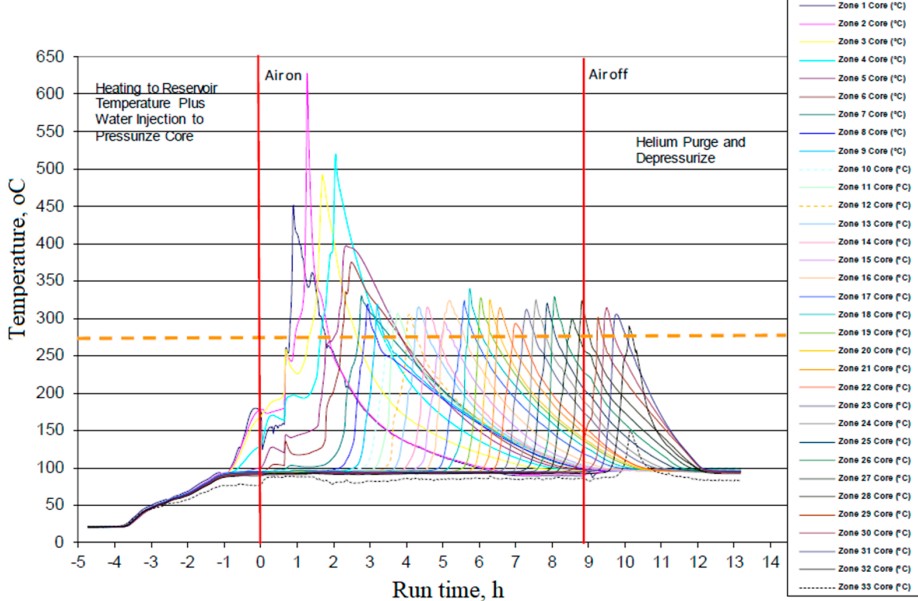

**Figure 3.** Forward Combustion Test Temperature Profiles (Color should be used).

The maximum peak temperatures for the first 6 zones were higher than the rest of the zones and were in the range of 375 to 626 °C. Despite the fact that the air injection was stopped after Zone 28 reached 324 °C, the following four zones (Zone 29, 30, 31, 32) demonstrated temperature levels similar to upstream zones due to continued combustion with stored oxygen. Their peak temperatures varied in the range of 290 to 314 °C. It should be noted that the operating pressure of 27.2 MPa is higher than the critical pressure of water.

The front velocity was calculated at the selected temperature of 275 °C, which is represented by the horizontal dashed line in Figure 3. The selection of the temperature level that is used to define the combustion velocity is based on the temperature range in which the oxidation reactions that are primarily responsible for mobilization of the oil occur. Generally, the temperature to track the front velocity exceeds 350 °C for heavy oils. In the case of light oils, mobilization of the oil is primarily associated with the combustion/oxidation reactions that occur in low-temperature range. The time at which zone attained this temperature was plotted against the corresponding thermocouple location for Zones 3 to 32 in Figure 4.

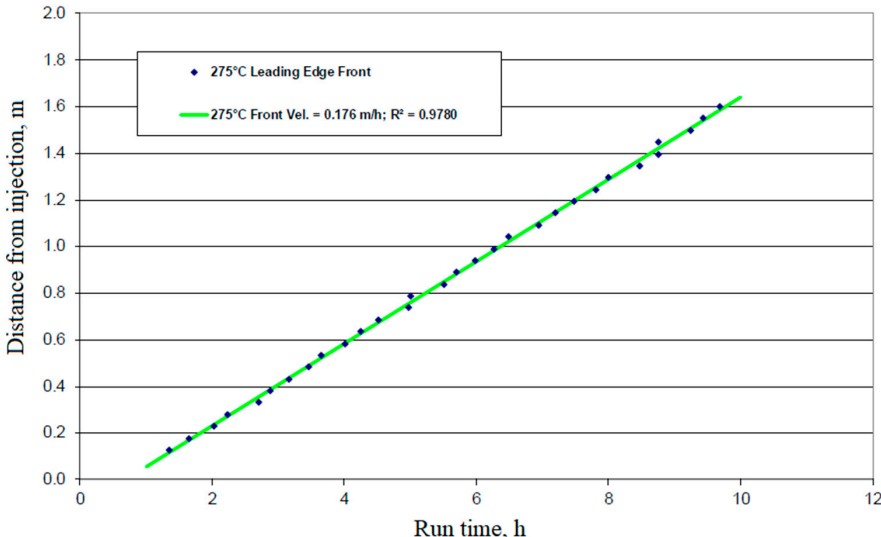

**Figure 4.** Forward Combustion Front Locations.

The slope of this plot gives the 275 °C front velocity at the air injection flux used in the test. According to the given slope of the front location, the advancement rate of the leading edge was 0.176 m/h at an air flux of 40.4 $m^3(ST)/m^2$ h. Following Figure 3, the front velocity would not change significantly for the 290–330 °C front temperatures at the abovementioned air flux.

The production of main combustion gases such as oxygen, nitrogen, carbon dioxide, and carbon monoxide as a function of runtime for the entire test period is given in Figure 5. During the first 2.5 h after the start of air injection, all the produced gases were collected inside a 3-L trap; no continuous gas stream was sent to the gas chromatographs and accidentally vented out; thus, only a small fraction of its residue was analyzed.

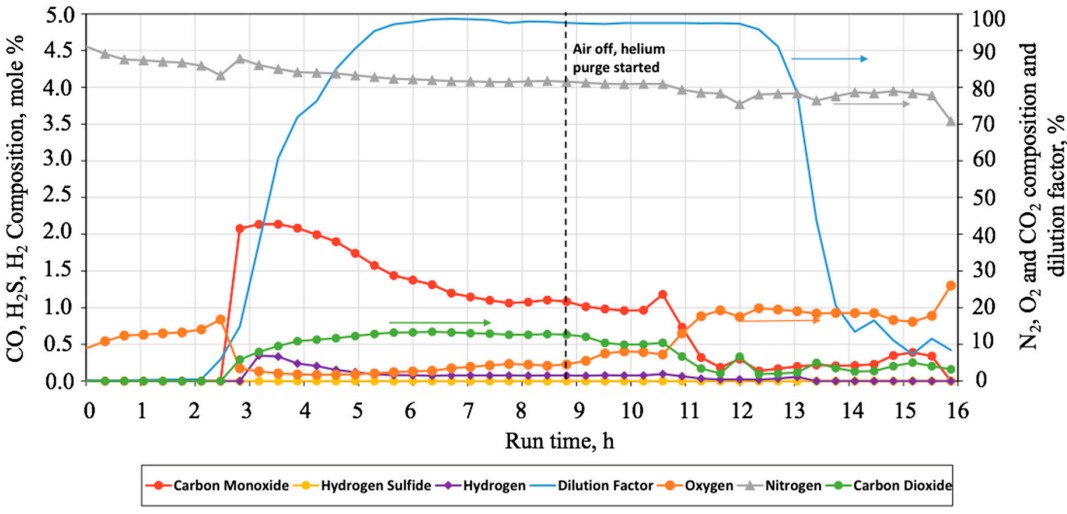

**Figure 5.** Produced Combustion Gas Compositions for Forward Combustion Test (Color should be used).

As can be seen in Figure 5, there was a steady production of carbon dioxide at about 13%, which indicates favorable bond–scission-type reactions. This was confirmed by good burning characteristics. Since the observed temperatures were not very high, the level of carbon dioxide was not attributed to the decomposition of the carbonate core [31]. According to the experience, the combustion gas composition results from the laboratory experiments correspond well with field-scale observations. The overall oxygen utilization was 61.5%, and unconsumed oxygen partially due to unburned oxygen

produced throughout the air injection period, but primarily due to stored oxygen in the burned section of the combustion tube test. It was later displaced during a helium purge which resulted in an oxygen peak in Figure 5. The overall apparent atomic hydrogen to carbon (H/C) ratio was 1.22.

Figure 6 below shows the cumulative production of oil and water over time.

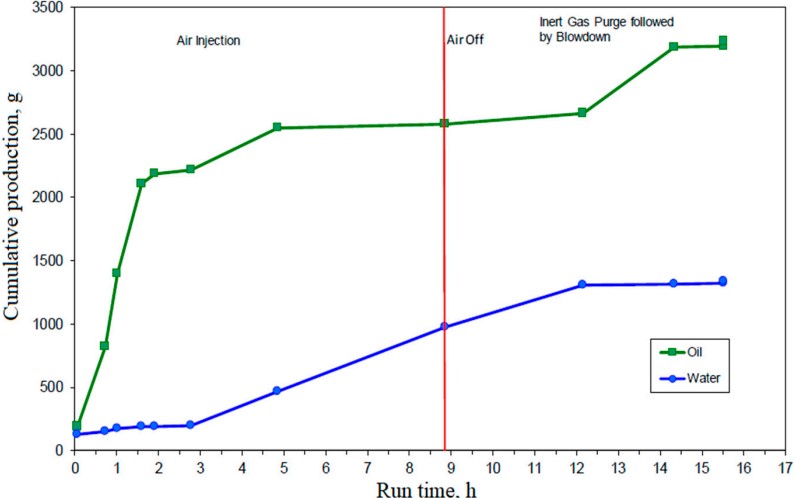

**Figure 6.** Forward Combustion Oil and Water Production Cumulative Masses (Color should be used).

According to the experiment, 3236.5 g of oil were produced, which corresponds to a value of recovery coefficient equal 91.4% taking into account the very small amount of initial oil in lines. Of the 3533 g of oil initially contained in the system, 91.4% was produced as liquids, 4.4% was consumed as fuel, 0.4% was produced as fuel gas, and 2.4% remained as residual in the core.

*3.2. Reverse Combustion*

The purpose of this test was to investigate the in situ combustion behavior of the restored-state core in a reverse combustion mode. Figure 7 presents the centerline temperatures as measured by the 33 thermocouples.

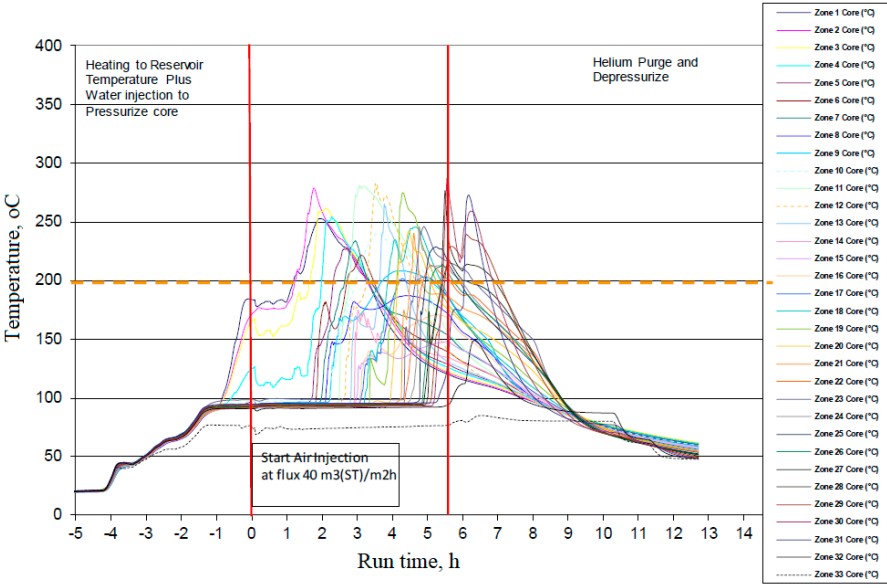

**Figure 7.** Reverse Combustion Test Temperature Profiles (Color should be used).

The heating of the three ignition zones also started at −0.9 h. The maximum peak temperatures for the first four zones are above 250 °C, while the subsequent Zones 5–10 peak at temperatures less than 250 °C. The following zones under numbers 11, 12, and 13 again exceed 250 °C peaks, while the next three zones remain below 200 °C. Mid zones demonstrated relatively low-temperature peaks and, from Zone 17, they started to increase, achieving the highest level at 288 °C at Zone 30. Lower peak temperatures can be explained by the kinetics of the reactions occurring during reverse combustion. The air injection was terminated after 5.6 h after the start of air injection and helium was injected when the leading edge of the high-temperature front reached Zone 30.

The front velocity for the reverse combustion was calculated at the selected 200 °C, which is represented by the horizontal dashed line in Figure 7. The time when each zone attained this temperature was plotted against the corresponding thermocouple location for zones 3 to 30, as seen in Figure 8. Two distinct stable periods were observed during the reverse combustion test (see Figure 8). These two sections were considered as stabilized combustion zones. The first stable section corresponds to the first seven Zones with relatively high peak temperatures. As was described earlier, there was a temperature peak drop in the mid zones, then an increase started from Zone 17, which resulted in a steeper front velocity slope.

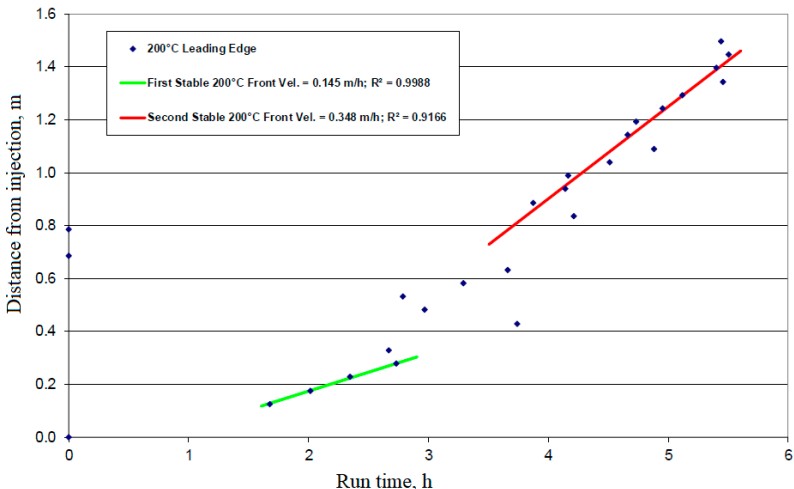

**Figure 8.** Reverse Combustion Front Locations (Color should be used).

Based on Figure 8, the advancement rate of the 200 °C leading edge at an air flux of 40.4 $m^3$(ST)/$m^2$ h was 0.145 m/h for the period 1.67 to 2.73 h, and 0.348 m/h for the period 3.87 to 5.50 h. Between 2.8 and 4.0 h, the advance of the front was unstable, with low temperature (<200 °C) peaks. The product gas concentrations as a function of runtime are presented in Figure 9.

For the typical sandstone combustion test in the high-temperature (bond–scission) mode, $CO_2$ concentration is 12–15%, and CO is 1.0 to 3.0%. For these carbonate combustion tests, similar levels were observed, although the levels of CO during the reverse combustion test were higher, possibly due to less stable combustion characteristics.

The production of the main combustion gases—oxygen, nitrogen, carbon dioxide, and carbon monoxide—is displayed in Figure 9. No measurable hydrocarbon was produced during the first 1.5 h after the start of air injection, only trace quantities of oxygen and nitrogen, slightly diluted by helium. Ignition was observed at 1.13 h; the first traces of carbon dioxide and light hydrocarbon were detected by the gas chromatograph at around 1.6 h.

The level of carbon dioxide production remained between 6 to 8% for the first 7 h, which indicates the gas production during the air injection period. However, typical favorable conditions for bond–scission-type reactions consistent with favorable burning characteristics normally result in carbon dioxide production at levels of 12–15%. In combustion tube tests on carbonate cores that exceed 500 °C (typical of heavy

oil combustion) or where water co-injection is used (e.g., wet combustion), $CO_2$ level exceeding 16%, and sometimes reaching 30% have been observed due to the decomposition of the carbonate core material. In the reverse combustion test, temperatures did not exceed much more than 300 °C, resulting in a lower level of $CO_2$ (see Figure 9).

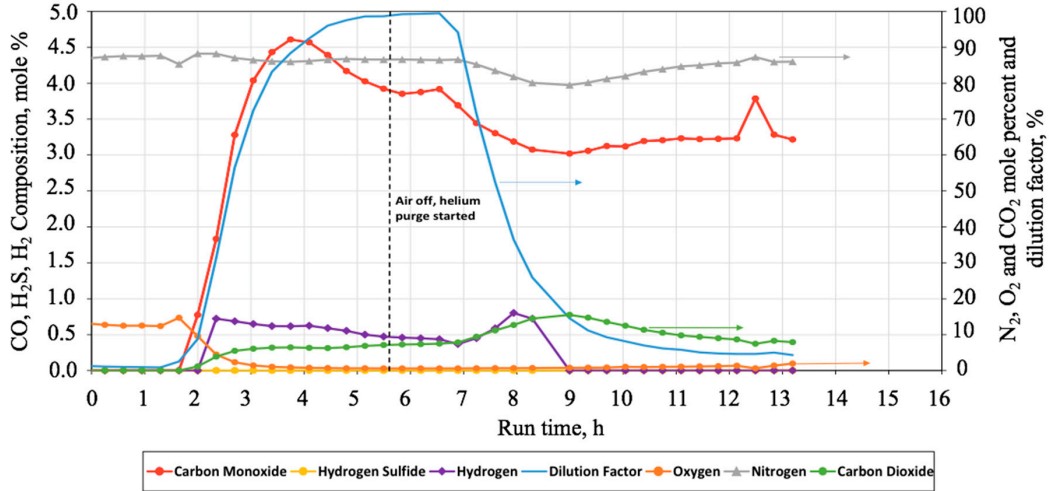

**Figure 9.** Reverse Combustion Produced Combustion Gas Compositions (Color should be used).

Nevertheless, oxygen consumption was nearly complete, indicating reactions with oxygen consumption but the without production of carbon oxides. It can be explained by water formation or liquid phase hydrocarbon oxidation. The unconsumed oxygen and the stored oxygen in the burned section were displaced during helium purge and appeared as an oxygen peak during the depressurization (see Figure 9). The overall apparent atomic hydrogen to carbon (H/C) ratio was 3.9, which is considerably higher than usual the 1.2 in the forward combustion test. It indicates oxygen addition reactions between the injected air and the significant quantity of warm residual oil in the core pack. This feature is one of the less attractive features of the reverse combustion. Figure 10 presents the cumulative liquid production over time.

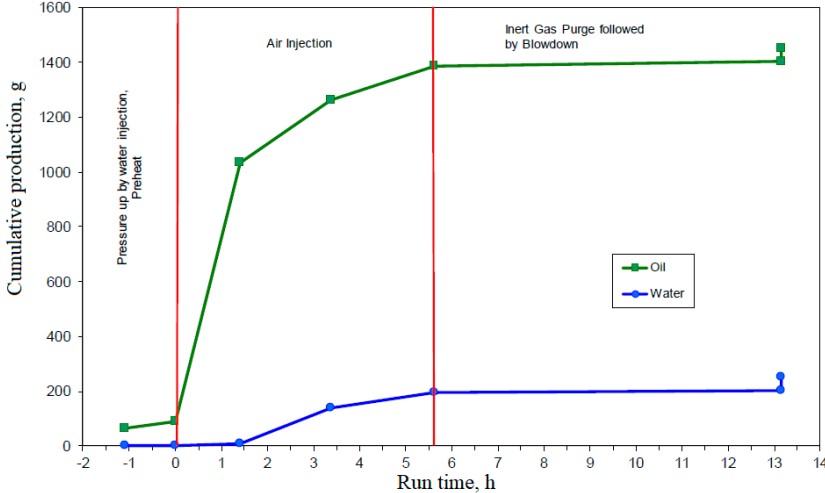

**Figure 10.** Reverse Combustion Oil and Water Production cumulative masses (Color should be used).

Oil production amounted to 1451 g including lines, which gives 42.5% recovery of the OOIP. A total of 4% was consumed as fuel, another 1% was consumed as fuel gas and 50% remained as residual on the core in the one-dimensional reverse combustion tube experiment. The initial water in

the system was 1596 g; 250.8 g was produced as a liquid, 25.2 g was produced as gas, and 1361.4 g remained as residual water.

## 4. Discussion

Combustion tube tests were performed to assess the suitability and potential of the selected oil reservoir for the implementation of an air injection-based EOR. Additionally, they can provide useful information regarding the combustion characteristics of the studied rock/oil system. These parameters are influenced by a wide range of factors, such as properties of the fluid, experimental pressure and temperatures, permeability, porosity, and composition of the rock matrix.

The maximum peak temperature for the forward combustion test was 626 °C while, in reverse combustion, the maximum recorded temperature was only 288 °C. The average peak temperatures are generally a function of the air flux and dependent on heat loss, thus, should be the subject of further studies during the numerical simulation.

Table 2 provides a summary of stabilized combustion parameters for both tests.

- Relatively low hydrogen to carbon (H/C) ratio of forward combustion indicates the high degree of high-temperature oxidation (HTO) occurring in the tube-pack. The overall apparent atomic H/C ratio for reverse combustion was 3.91 which is higher than usual (H/C is 1.2 in the forward combustion test). It indicates the occurrence of oxygen addition reactions between the injected air (oxygen) and the significant quantity of warm, residual oil in the core pack. This is one of the less attractive features of reverse combustion.
- The combustion front velocity was 0.176 m/h for the forward combustion test. Reverse combustion demonstrated two distinct velocity periods with the front velocity of 0.145 m/h at the first stage and 0.348 m/h at the second based in the produced carbon dioxide. These two sections were considered as stabilized combustion zones. Similarly to the peak temperatures, the front velocity is affected by air flux and heat loss decreases its value. The combustion front development and front velocities are crucial for prediction on field-scale performance.
- The recovery efficiency can be used for making an economic projection of field performance. Oil recovery for the forward combustion was as high as 91.4% of the initial oil in place with 2.4% remaining residual, while only 43% was produced as liquids during reverse combustion process with 50% remained as residual on the core. This result can be explained by the API values of the oil samples. Forward combustion has a wide range of oils from 10 to 40 °API, while for the reverse combustion 5 to 20 °API considered to be favorable [17] This parameter is also a subject of "history matching".
- According to the results (see Table 2) reverse combustion required a higher amount of air at the first stable section than during forward combustion. However, when the temperatures started to increase again and the front velocity slope became steeper the air requirement decreased sharply. The overall air requirement was 253.5 and 146.9 $m^3(ST)/m^3$ for forward and reverse experiments, respectively. The air requirements determine the compression capacity affecting the overall project economics.

In contrast with sandstone reservoirs, in carbonate reservoirs resulting in 12–15% $CO_2$ concentration and 1.0 to 3.0% CO, there are reactions other than HTO, LTO, but also carbonate decomposition with products of reaction as $CO_2$, CO, $O_2$, $N_2$, and water [31]. Generally, during heavy oil combustion tests on a carbonate core exceeding 500 °C or wet combustion, $CO_2$ might be in the range of 16–30% due to the decomposition of the carbonate core material. According to some of the literature, the decomposition reaction of carbonates [32]. is assumed to take place at temperatures above 700 °C [33] Thus, at the given maximum temperature (300 °C), the contribution of dolomite and calcite can be insignificant, similarly to [34].

Dependence on initial temperatures was not evaluated within these experiments. However, it might affect the peak temperatures and the combustion-zone velocities. Both experimental and

numerical tests should focus on the determination of the kinetic parameters and chemical reactions adequately describing the processes.

**Table 2.** Summary of stabilized combustion parameters.

| | Forward Combustion | Reverse Combustion | |
|---|---|---|---|
| Combustion front, °C Leading edge | 275 | 200 | |
| Time interval by velocity, h | 1.35 to 10.11 | 1.67 to 2.73 | 3.87 to 5.5 |
| Gas chromatograph interval, h | 4.23 to 9.17 | 3.1 to 4.5 | 5.3 to 7.1 |
| Air fuel ratio, $m^3$(ST)/kg | 10.84 | 13.52 | 13.38 |
| Combustion front velocity, m/h | 0.176 | 0.145 | 0.348 |
| Air required, $m^3$(ST)/kg | 229.5 | 279.49 | 116.08 |
| Fuel required, $kg/m^3$ | 17.82 | 19.97 | 8.45 |
| Apparent Atomic H/C ratio | 1.45 | 5.10 | 4.69 |
| The percent Oxygen Utilization, % | 84.2 | 96.62 | 97.37 |
| The percent conversion of reacted $O_2$ to carbon oxides | 72.4 | 38.37 | 41.33 |
| ($CO_2$ + CO)/CO Ratio | 11.13 | 2.43 | 2.87 |
| ($CO_2$ + CO)$N_2$ Ratio | 0.17 | 0.12 | 0.13 |
| Mole Percent $O_2$, % | 21.28 | 21.04 | |
| $N_2/O_2$ Ratio | 3.69 | 3.75 | |

## 5. Conclusions

The work was conducted to study the combustion behavior of the oil sample from the target field and evaluate its burning characteristics, incremental production of oil, and water, air, and fuel requirements. Forward and unique reverse ISC combustion methods were examined to predict feasibility for their application in the target oil field.

- HPCT tests on a 100-mm diameter high-pressure combustion tube laboratory tests using actual reservoir samples were performed, at a pressure of 27 MPa and an air injection flux of 40 $m^3$(ST)/$m^2$ h at an ignition temperature of 175 °C.
- Favorable test results were confirmed by the propagation of a steady combustion front through the core pack and a stable product gas composition for both tests.
- The oil recovery was 91.4% for the forward combustion and 43% for the reverse combustion tests. For the forward combustion of the 3533 g of oil initially in the system, the above mentioned 91.4% was produced as liquids, 4.4% was consumed as fuel, 0.4% was produced as fuel gas and 2.4% remained as residual. Similarly, for reverse combustion, 43% was produced as liquids, 4% was consumed as fuel, 1% was produced as fuel gas and 50% remained as residual on the core in the one-dimensional reverse combustion tube experiment.

There are factors affecting the overall performance of the experiments, such as air flux, heat losses, initial temperatures, and initial water/oil saturation. While the test configurations employed were selected to minimize the number of equipment parameters that were changed, a future combustion tube test of reverse combustion, with oil production in a downward direction, would provide valuable insights. The orientation of the combustion tube test during reverse combustion, permeability values, the effect of carbon decomposition, and other factors affecting the performance of reverse combustion should be examined further during additional laboratory tests and numerical investigations with the implementation of the chemical model.

Reverse combustion can be used as a preheating method before steam flooding or other EOR technique. The initial oil saturation of the given reservoir was comparatively high and viscosity of initial oil also was lowered during reverse combustion, thus forward combustion could be performed

further to achieve a higher oil recovery. Reverse combustion pre-treatment can lead to the development of highly permeable paths between wells. The reverse combustion tests on oil samples with API in the range of 5 to 20 °API [17], in comparison with the oil sample presented in this research, also could reveal more insights about the reverse combustion process.

Under the given conditions, the forward combustion process demonstrated better performance and was more efficient at mobilizing oil from the core pack in comparison with the reverse combustion test. However, experiments conducted in this study are not enough to declare the higher efficiency of the forward combustion method. Generally, this method is more technically developed and demonstrated higher recovery factors. Nevertheless, forward combustion mode has viscosity limitations. Reverse combustion, in its turn, can be applicable for very heavy crude oil, low-permeability reservoirs, and can serve as a preheating method. Meanwhile, there is a probability of spontaneous ignition, combustion instabilities, possibility of shifting to forward mode. As was already mentioned, the reverse combustion tube test with air injection in a top-down direction identical to forward combustion would be useful. Nonetheless, this research provides an important set of experimental data obtained at the reservoir conditions that would be encountered in the field in the domain of increased-pressure operation. Both methods have high-cost air compression and risks associated with oxygen breakthrough. Thus, it is crucial to conduct the numerical modeling of the experiments, further validate the numerical models against experimental results, and perform the field-scale modeling to predict the performance of both methods. Additionally, this process will allow the determination of favorable conditions where reverse combustion can be successfully applied. Reverse combustion must be further studied using different oil and core samples. Further numerical simulation of reverse combustion experiment can reproduce the possible combustion channels in response to different operational variables and heterogeneities in the permeability.

**Author Contributions:** Conceptualization, A.A., E.P., and M.U.; methodology, S.M., G.M., M.U.; formal analysis, S.M., G.M., M.U., E.P., A.C. and A.A.; investigation, E.P. and A.A.; resources, S.M.; writing—original draft preparation, A.A.; writing—review and editing, A.C., S.M., M.U., E.P., A.A.; visualization, M.U., A.A., E.P.; supervision, A.C., S.M., G.M.; project administration, A.C.; funding acquisition, S.M. All authors have read and agreed to the published version of the manuscript.

**Funding:** This research received no external funding.

**Acknowledgments:** The authors would like to acknowledge the researchers of Skoltech Integrated Center for Hydrocarbon Recovery and University of Calgary Schulich School of Engineering who helped to conduct the experiments.

**Conflicts of Interest:** The authors declare no conflict of interest.

## Abbreviations

The following abbreviations are used in this manuscript. ISC, in situ combustion; EOR, enhanced oil recovery; HPCT, high-pressure combustion tube; API, American Petroleum Institute; LTO, low-temperature oxidation; HTO, high-temperature oxidation; H/C, apparent atomic hydrogen to carbon ratio; OOIP, original oil-in-place.

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
