# Peer review of "Experimental Investigations of Forward and Reverse Combustion for Increasing Oil Recovery of a Real Oil Field"

_energies, doi:10.3390/en13174581_

Round 1
Reviewer 1 Report
This paper studies the combustion behavior of the oil sample in the target field and evaluates its combustion characteristics. It also measures the incremental production of oil and the requirements of water, air and fuel. The forward combustion and reverse combustion methods were studied to predict the feasibility of their application in the target oil field. However, this paper still has the following shortcomings.
- EOR appears in line34 for the first time, its full name needs to be written.
- The article mentions that there are many different technologies for underground combustion, several types need to be introduced to lead to the forward combustion and introduce the advantages of the forward combustion.
- The API at Line78 needs the full name and its meaning needs to be introduced.
- It is mentioned in the paper that the oil recovery rate of reverse combustion is much lower than that of forward combustion, and the front velocity of the fuel is high. So what the research value of reverse combustion is.
- The principle of the HPCT experimental setup needs to be introduced in detail.
- The different areas of the combustion tube need to be drawn in Figure 1.
- In line 311, the reason for the low oil recovery of reverse combustion needs to be re-explained.
Author Response
Please, see attachment

Reviewer 2 Report
This paper presents experimental results of forward and reverse combustion tube experiments to access the feasibility of the process in a light carbonate reservoir. I have the following questions and comments for the authors:
- To what extent were the original reservoir properties (such as the original porosity and permeability) preserved in the experiment? Carbonate reservoirs are known to have vugs and microfractures, resulting in high heterogeneity. Since the goal of the experimental study is to inform field implementation, the difference between the reservoir and the experimental parameters can be highly influential in the final conclusion.
- Permeability in the reverse combustion test is almost half compared to the permeability during the forward test, as noted in Table 1. Please explain why that was the case and to what extent this is affecting the final results.
- The reason for the significantly lower peak temperature during the reserve combustion was not explained.
- In the forward combustion air was injected top-down and production collected at the bottom, whereas during the reverse combustion, the air was injected at the bottom and production collected at the top. It appears to me that because of this set-up flow due to gravity will enhance the oil production in the forward combustion. Thus, the final conclusion that reverse combustion performs poorly can possibly be because of the experimental set-up. Could the authors explain how much gravity could be influencing the high performance during the forward test?
- The previous experimental studies by Perry et al. (1960) on reverse combustion also indicated about 50% recovery, similar to the presented work. How is the current study enhancing the previous knowledge?
- To what extent is the carbonate reservoir influencing the CO2 and CO concentration in the produced gases? Please compare it with a typical concentration in a sandstone.
- 6 and Fig. 10 show the oil and water production in the forward and reverse combustion experiments, respectively. However, a couple of things stand out – the forward combustion experiment lasted about 16 hrs. while the reverse lasted 13 hrs., the air was shut off at 9 hrs. in forward and at 5.5 hrs. in reverse (which means much longer air injection period in the forward). Can the authors explain the different experimental procedures followed and to what extent this is influencing the final conclusion?
- Why did the reverse combustion result in a much lower water production compared to the forward?
- The discussion section should include a better explanation of the results, beyond just stating what was observed during the experiment. This is important for any field implementation.
- In the conclusion section, it is mentioned that “this research gives important insights about reverse combustion and favorable conditions where it can be successful”. However, it is not clear how the experiments conducted in the study explore the favorable conditions in which reverse combustion will be successful.
- Some discussion should be included regarding how the different experimental parameters (H/C, recovery efficiency, etc.) will translate to the reservoir setting. One of my major concerns is the vertical set-up, which clearly introduces the effect of gravity in the overall performance seen in the experiments.
- The introduction section is a bit long. The advantages and disadvantages of forward versus reverse combustion should be clearly and concisely stated. Also, the novelty of the current experiments, compared to the previous work should be included.
Author Response
Please, see attachment

Round 2
Reviewer 1 Report
accept
Author Response
Thank you very much for your comments, we really appreciate them!
Reviewer 2 Report
Thank you for providing further clarification. I have some follow-up questions.
- It was noted that permeability variation does not have a significant impact on combustion, which I can partially agree with since it is a thermal process. However, since the goal here is to compare the performance of forward and reverse combustion processes for future field application, if the permeability for one experiment is half of the other, the conclusion will be misleading. Ultimately Darcy's equation still applies for the mobile oil and higher permeability is undoubtedly going to result in a higher flow rate. It is also not clear why the permeabilities were so different in the reverse and forward cases if a similar experimental procedure was applied.
- I do not agree with the authors that gravity is not playing a role. While the use of a vertical set up is not in question, the issue is that the oil production is downwards in one experiment and upwards in another and the conclusion that one produces more oil than another is partially influenced by the set-up itself. The bigger question is how this study can help future application of reverse combustion and in my view, the conclusion of this study is misleading as permeability and experimental set-up are both biased against it.
- The authors mention a few times that some of the factors (such as mineral deposition) will be studied more closely in the numerical simulation. However, numerical simulations have a lot more variables that can be tweaked to produce the desired outcome (especially in history matching). An experimental study is a more realistic demonstration of the various mechanisms including mineral deposition and therefore at a minimum they should be discussed alongside the experimental outcomes, rather than only evaluate them in the simulation (which may not be as realistic). Numerical simulations also do not exactly replicate the transient effects, and a more rigorous analysis of experimental results is more insightful, than the analysis of history match results.
- To the question of why reverse combustion results in a much lower water production, the authors have responded by just reiterating the observation. However, please explain from a physics or mechanistic standpoint why water is not mobilizing as much. In my view, drainage against gravity in the reverse combustion is hindering both oil and water production.
- The authors mention that "The main novelty is that there is no available data on reverse combustion experiments, that were performed under reservoir conditions (high-pressure)" and I agree with that. However this is also my main concern that the experimental process/set-up has introduced bias in the conclusions from this study that reverse combustion performs poorly compared to forward which is misleading for the audience.
Author Response
Please, see the attachment

Round 3
Reviewer 2 Report
Thank you for the revisions. Because of the limitations of the experimental set-up, it is not accurate to conclusively declare that forward combustion will perform better in the field than reverse combustion based on the experiments conducted. Thus, the authors may consider revising the lines in the manuscript that declares as such and mention clearly that the difference in performance is at least in parts due to the different conditions in the two experiments, so that it does not mislead the audience. For example:
- (line 16-17, please correct as conditions were not identical) "One forward and one reverse combustion tube tests were carried out using the high-pressure combustion tube (HPCT) experimental setup under identical conditions".
- (line 22-23, not accurate more work needs to be done for that conclusion) "It is defined that forward combustion has higher efficiency and can be used for the development of the oil field".
- Revise the conclusion section and refrain from declaring that forward is better for the field application only based on the experiments conducted in this study.
Author Response
Thank you very much for the comments and suggestions!
The file with answers is attached
